# Is Government Spending an Important Factor in Economic Growth? Nonlinear Cubic Quantile Nexus from Eastern Europe and Central Asia (EECA)

Ali Shaddady 

Department of Finance, King Abdulaziz University, P.O. Box 80200, Jeddah 21589, Saudi Arabia; amshaddady@kau.edu.sa; Tel.: +966-126400000

**Abstract:** This study rigorously investigates the non-monotonic phenomenon of the government spending–growth nexus in the EECA. Using panel data from 19 countries over the period 1995–2019, a nonlinear quadratic estimator and cubic nonlinear estimator were applied to quantile regressions. The preliminary findings revealed a negative linear nexus of government spending and economic growth using a linear model, while the nonlinear models (i.e., quadratic and cubic nonlinear estimators) indicated evidence of nonlinearity in the nexus in the EECA over the study period. Furthermore, the study found strong evidence of the existence of an inverted "N-shaped" (nonlinear cubic) nexus between government spending and growth, which can be interpreted as the typical credit-driven boom-and-bust cycle in most EECA countries. Finally, in elucidating the nexus between government spending and economic growth, the study found that most macroeconomic and governance variables are relaxing in explaining GS.

**Keywords:** government spending; economic growth; nonlinear cubic; quantile approach; nonlinear quadratic



## 1. Introduction

Throughout the history of the world, economies have faced episodes of fiscal crisis attributable to the impacts of government expenditure on economic growth. Fundamentally, this is because of governments' incapacity to bridge deficits between state spending and revenues and the tendency toward large and growing governments, which can have deleterious effects on national economic growth. The typical economic prescription to address deficits is to reduce state spending by cutting budgets and curbing the growth of general government spending that outpaces output via fiscal consolidation programs to control public expenditures, which are often a condition of lending in countries that face financial deficits and debt burdens. Parallel to this thrust, there is a call for a "fiscal space" in which governments can invest more in providing productive public goods that can foster economic growth, particularly over the long term (Divino et al. 2020).

Clearly, economic analyses and debates on this issue have implications for political economy and policy, and theoretical and empirical studies have devoted considerable analysis to government spending's positive and negative influences on economic growth. On the positive side, increasing investment in core economic areas, such as physical and institutional infrastructure, is assumed to be conducive to economic growth (Romp and De Haan 2007; Nurudeen and Usman 2010; Ravn et al. 2012). Conversely, as government spending increases, distortionary effects of high public borrowing and taxes, bureaucratic costs, and diminishing returns on public capital become more prevalent effects, which undermine productivity and economic growth (Fedderke et al. 2006; Hajamini and Falahi 2018).

In summary, these opposing views introduce the non-monotonic nexus of government spending in economic development. While the existence of this relation between government spending and economic growth has been widely attested (Asimakopoulos

and Karavias 2016; Kim et al. 2018; Atems 2019), its shape and features remain ambiguous, leading many economists to pose questions. One issue of concern is "multiple steady-state levels", whereby the performance of governments' fiscal operations may depend upon other institutional or macroeconomic dimensions (Olaoye et al. 2020). Several factors dominate the nonlinear nexus and raise debates over government spending's influences on economic growth, including: (1) institutional or macroeconomic dimensions, such as macroeconomic shocks and institutional infrastructure; (2) margin of government expenditure or valid tipping point (i.e., threshold), after which additional spending precipitates economic slowing; (3) asymmetric information, attributable to a lack of transparency or weak institutional frameworks in certain economies; (4) financing behavior that predicts nonlinearity in the government spending–growth nexus, attributable to "tax-financed" government spending; and (5) shortcomings of pooling countries (i.e., both developed and developing) that fail to manage region-specific effects and heterogeneity across regions.

Against this backdrop, this study attempts to provide a scholarly contribution to the extant literature in four ways. Firstly, this study rigorously investigates the heterogeneity in the estimated relation between government spending and economic growth (i.e., the non-monotonic nexus) by re-examining the shape of the non-linear nexus between government spending and economic growth. Secondly, and most importantly, the study adopts the cubic quantile method to reconcile the mixed results in the literature about the relation between government spending and economic growth and uses the instrumental variables method and generalized method of moments estimation (GMM) to address the possibility of a potential endogeneity issue. Thirdly, to address the issue of region-specific effects identified by some empirical studies, and the existence of heterogeneity across regions (Ghirmay 2004; Odhiambo 2008), which can provide misleading results, this study focuses on Eastern Europe and Central Asia (EECA).[1]

The EECA region has experienced a tumultuous recent history, during which many of its states have transformed from Soviet Communism to free market economies over recent decades, spurred by massive international and domestic investments (Gray et al. 2007), to the extent that it is now considered a middle-income region.[2] Consequently, this study's outcomes can be generalized to other middle-income economies. The EECA countries share common geo-economic characteristics. While the 19 constituent EECA economies studied in this research share similar macroeconomic and institutional features as the other emerging economies, they are distinguished by the economic outcomes they experience, which is the rationale for the notable economic research interest in the economies of Eastern Europe and Central Asia (Poznańska and Poznański 2015). Finally, this study examines whether macroeconomic and macro-governance factors serve as substitutes or complements in shaping the government spending–growth nexus.

The empirical work reveals fresh evidence that supports the cubic nonlinear nexus between government spending and economic growth in the EECA and provides strong evidence of the existence of an inverted "N-shaped" relation. Furthermore, the study estimates the optimal level of government spending and determines its effects on growth when it is below or above the threshold level. This finding may have important practical policy implications, particularly for policymakers and governments in developing and middle-income countries. Governments should be discreet about crowding-out effects that may be operating because of increasing government spending and weak governance with regard to institutional accountability.

The remainder of this paper is organized as follows. Section 2 presents the theoretical background and literature. Section 3 describes the methodology, variables, and data; the empirical results are presented in Section 4, which also provides validity and robustness checks. Finally, Section 5 offers concluding remarks.

## 2. Theoretical Background and Literature Review

### 2.1. Brief Theoretical Background

The government spending–economic growth nexus remains a source of debate in both the theoretical and empirical literature (Arpaia and Turrini 2007). The potential to use government spending as an instrument to enhance growth has been criticized theoretically, while there are other opinions that emphasize government spending's function and the public sector's capacity to stimulate and administer economic development (Stoilova 2017). Consequently, to attain a better theoretical understanding of this debate, it is necessary to focus on two major elements: the sustainability of public finance, which can help evaluate the influence of government spending; and the development of a benchmark to help assess the role of spending and fiscal policies in overall growth (Arpaia and Turrini 2007).

There is consensus among most economists that a certain degree of government spending is a major and necessary component of economic and national income; debates about public spending primarily concern the level at which this becomes a counterproductive and onerous burden on the private sector (Nuru and Gereziher 2021). Government spending can enhance aggregate output significantly (Solow 1956; Cass 1965; Romp and De Haan 2007; Nurudeen and Usman 2010), but at the expense of adverse consequences attributable to financing via increased taxes and borrowing (Romer 1986; Lucas 1988; Becker et al. 1990; Rebelo 1992; Fedderke et al. 2006). However, empirical evidence remains inconclusive about the fundamental relation between public finance and growth theory—i.e., whether increasing government spending enhances economic growth (Alshahrani and Alsadiq 2014).

Theoretically speaking, Keynesian macroeconomic theory supports the role of government spending in enhancing economic growth by increasing aggregate demand, production, and national income, which are posited to enhance rapid economic development (Larch and Lechthaler 2013). Nevertheless, the neo-classical school argues that increased government spending attributable to raising taxes or borrowing may slow economic growth, because it crowds out the private sector, reduces consumption, and decreases real wages and production (Hajamini and Falahi 2018). This theoretical debate focuses on the ways in which real economies operate, particularly producers and consumers, and the way they react to public finance policy behavior (i.e., political decisions), in association with numerous interrelated economic and macro-governance indicators. Furthermore, there is potential information asymmetry in the relation between government spending and economic growth. These factors may lead to the major consequence of nonlinearities in the nexus between government spending and economic growth (Olaoye et al. 2020).

### 2.2. Literature Review

#### 2.2.1. Nexus between Government Spending and Economic Growth

The efficacy of government actions in stimulating or impeding economic growth hinges on fiscal policies (Auerbach and Gorodnichenko 2012). Government spending's influence on economic growth is hotly debated in various contexts, as explained previously, in both theoretical and empirical studies (Zungu et al. 2020). Very broadly, the empirical findings on the government spending–economic growth nexus can be categorized into the following groups according to their outcomes.

The first set of studies concluded that there is a negative correlation between government spending and economic growth, largely because of government spending's role in exacerbating inflationary pressure and the crowding-out effect (Barro 1990; Nelson and Singh 1994; Agell et al. 1997; Gemmell and Au 2013; Onifade et al. 2020). The crowding-out effect occurs when increased public sector spending reduces or even eliminates private sector spending, or when state expenditure is financed by increased taxes, which may also discourage private sector investment (Afonso and Sousa 2011). Moreover, expansionary government spending policies with excessive financing through borrowing can reduce private-sector confidence, because of the inevitability of higher taxes to serve public debts, which has a detrimental influence on productivity and economic growth, particularly in the long run (Shonchoy 2010). Sawyer (2012) pointed out that financing government

spending inhibits long-term economic growth because of unsustainable debt and decreases the private sector's confidence (and investment) in the short- and medium-term.

The second body of research includes studies that found a positive correlation between government spending and economic growth (Ghali 1999; Wu et al. 2010; Ghose and Das 2013; Akpan and Abang 2013; Kimaro et al. 2017). Typically, the proponents of this approach extol government expenditure's critical role in harmonizing conflicts between social and private interests by providing the optimal social direction for development and growth. Ghali (1999) indicated that high public sector investment could lead factor and product markets to work more efficiently and produce substantial spillover effects on economic and private sector growth. Similarly, Akpan and Abang (2013) pointed out that enhanced public investment in socioeconomic and physical infrastructure increases economic growth. Consequently, high government spending on education, health, and infrastructure (e.g., communications, roads, and power) is presumed to enhance labor productivity, increase private sector investment, reduce production costs, and increase latent national economic resources (e.g., the quality of human resources and efficiency of transportation), all of which contribute positively to economic growth. Moreover, Prasetyo and Zuhdi (2013) emphasized government spending's significant contribution in enhancing national economic growth by maximizing the efficiency of resource allocation, which is supported by the high performance attributable to government accountability (Kimaro et al. 2017).

The third group of studies proposes the hypothesis that there is a non-linear nexus between government spending and economic growth, and numerous studies have illustrated a non-monotonic relation between government spending and economic growth (Pevcin 2004; Chen and Lee 2005; Mavrov 2007; Aydin et al. 2016; Iyidogan and Turan 2017; Olaoye et al. 2020). However, there is a strong paradox in explaining the nature of government spending's influence on economic growth. The mixed positive and negative effects in most studies have tended to demonstrate nonlinear (quadratic) relations, as the quadratic nexus was used to explore the optimal level of government spending and its influences on economic growth levels. Barro (1990) demonstrated this nonlinear causality (normally distributed relation), whereby the dominating role of government spending raises the marginal productivity of capital, exerting a positive influence on economic growth; however, an increase in government spending above an optimal level (by raising taxes) reduces economic growth through disincentive effects. Heitger (2001) stressed that increased government spending should enhance economic growth when the government is investing in core public goods; however, this positive effect may tend to reverse if the government invests public money in private goods, and thereby crowds out the private sector.

Olaoye et al. (2020) adduced several reasons for the non-monotonic relation among government spending and economic growth. First, macroeconomic elements may cause this nexus, in which case macroeconomic shocks would influence the relation (Kim et al. 2018; Olaoye et al. 2019; Olaoye and Aderajo 2020). Second, several studies have posited that government expenditure creates a valid tipping point or threshold, beyond which it actively impedes economic growth. This approach supports a normally distributed nexus (Forte and Magazzino 2016; Hajamini and Falahi 2018). Third, government expenditure can exhibit an asymmetric information structure, particularly when there is no perfect information about government fiscal operation available, together with weak governance, obstructive levels of bureaucracy, and a lack of transparency (Hung and Lee 2010; Paleologou 2013). Fourth, business cycle indicators may exhibit asymmetric behavior, and associated asymmetries can translate into government expenditure and economic growth (Chen 2014; Combes et al. 2017). Consequently, for these reasons, the assumption of a linear nexus between government spending and economic growth is relaxed, while it is assumed that a nonlinear nexus is more efficient in light of the existence of the non-monotonic relation in other institutional or macroeconomic factors and government spending. Nevertheless, the shape of this nonlinear relation remains ambiguous and needs to be explored further.

2.2.2. Nexus between Macro-Governance Indicators and Economic Growth

Traditionally, economic growth research has concentrated on growth's relation with other economic indicators, but increasing attention is now being given to investigating the effect of non-economic variables, such as macro-governance indicators (Huang and Ho 2017; Erdoğan et al. 2020). Indeed, macro-governance dimensions clearly play a key role in growth, and different governance framework executions within the same systems affect the variety or variation in economic performance, but the nature of the relations among particular governance practices and economic performance remains unclear (Grindle 2004).

Numerous empirical studies have investigated whether macro-governance dimensions, particularly those based upon the World Bank's Worldwide Governance Indicators (WGIs), are beneficial to economic growth and performance. For instance, Kaufmann et al. (1999) reported that WGIs affect economic growth significantly. More generally, studies have demonstrated that economic growth is facilitated by a good macro-governance framework (Dollar and Kraay 2002), rule of law (Rigobon and Rodrik 2005), regulatory quality associated with enhanced foreign investment and trade (De Groot et al. 2004), government effectiveness (Jalilian et al. 2007), and political stability (Huynh and Jacho-Chávez 2009). Some studies have explored multiple WGIs' effects on economic growth simultaneously, including Méndez-Picazo et al. (2012), who reported positive effects of four WGIs (government effectiveness, voice and accountability, control of corruption, and rule of law), and Fayissa and Nsiah (2013), who emphasized the positive significant effects of six WGI dimensions (voice and accountability, regulatory quality, rule of law, political stability, government effectiveness, and control of corruption). Thus, the literature exhibits consensus that WGIs' actual and potential influences differ across the different dimensions of governance.

Improvement in macro-governance dimensions enhances national economic growth by attracting foreign investment and trade in addition to reducing economic crises' adverse effects and improving people's quality of life (Huang and Ho 2017). Accordingly, this study follows in this vein by proposing a positive hypothetical nexus between governance dimensions and economic growth.

2.2.3. Nexus between Other Macroeconomic Indicators and Economic Growth

Certain macroeconomic indicators, including foreign direct investment (FDI) (Shahbaz et al. 2022), inflation, interest rate, and effective exchange rate, are known to influence the nexus between government spending and economic growth (Van Dan and Binh 2019). Indeed, the importance of a sound macroeconomic environment is equal to that of a good macro-governance environment, not only because of sound macroeconomics' function in promoting growth, but also because the macroeconomic environment's contribution, coupled with a good governance environment, drives economic growth, which allows effective and credible policy shifts, creates effective government spending, and potentially reduces government spending shocks (Pradhan et al. 2015).

The relation between macroeconomic variables and economic growth in times of economic stimulus or government spending shock is inconclusive (Ulucak 2019). The neo-Keynesian approach posits that increased government spending may appreciate the real exchange rate, and thus impede economic growth (Chen and Liu 2018). In contrast, Miyamoto et al. (2019) pointed out that a positive government spending shock will depreciate the real exchange rate, and thereby enhance economic growth. The expected outcome of this research with regard to the relation between macroeconomic variables and economic growth based upon existing literature may be either negative or positive in this nexus. Increased government spending can create high inflationary pressures and impede economic growth because of increased production costs (especially with regard to labor) (Haberler and Salerno 2017; Mandeya and Ho 2021). This view emphasizes inflation's adverse effect on economic growth, as proposed in this research. Conversely, some studies argue for a positive correlation between FDI and economic growth, which is attributable to FDI being attracted when government spending expansion is accompanied by tax concessions

and relaxed protectionist policies, which can reinforce economic growth with additional foreign capital, technology, and high aggregate productivity, particularly for developing and emerging economies (Osei and Kim 2020). In the context of the real interest rate, this study assumes a negative relation between a high interest rate and economic growth, because there is an inverse nexus between the two: lower interest rates create steady and stable economic growth, while higher interest rates lead to a decline in investments, which impedes economic growth (Shaukat et al. 2019).

## 3. Materials and Methods

### 3.1. Methodology

The model proposed in this study consists of three equations estimated jointly. The first is a conventional linear formulation that examines the share of government spending over economic growth, augmented by macro-economic variables coupled with macro-governance variables. The second expresses the non-linear relation by introducing a quadratic function in the government spending variable, such that the direction of the variable indicates the shape of the non-linear quadratic nexus between government spending and economic growth, in which a "... positive direction indicates a U-shaped nexus; conversely, the negative direction indicates an inverse U-shaped nexus" (Alnori 2020). The third equation is a cubic formulation of government spending, in which the consistent sign between government spending and government spending cubed indicates a non-linear cubic relation, while a positive sign for both variables indicates an N-shaped non-linear nexus, and a negative sign indicates an inverse N-shaped relation between government spending and economic growth ((Shahbaz et al. 2019).

The estimates are constructed based upon unbalanced panel regression by building on existing empirical models to reveal government spending's potential effect on economic growth using pooled ordinary least squares (POLS) (Beckmann et al. 2016; Sekrafi and Sghaier 2018; Linh et al. 2019; Miniesy and AbdelKarim 2021). Specifically, the linear regression model is presented as:

$$EG_{it} = \beta_0 + \beta_1\, \boldsymbol{GS_{it}} + \beta_2 FDI_{it}^2 + \beta_3\, INF_{it} + \beta_4\, EXR_{it} + \beta_5\, IRR_{it}$$
$$+ \beta_6 VA_{it} + \beta_7\, PS_{it} + \beta_8\, GS_{it} + \beta_9\, RQ_{it} + a_i + y_t + \varepsilon_{it} \tag{1}$$

in which $i$ is country; $t$ is time; $EG$ is the dependent variable (economic growth using real GDP growth rate as a proxy); $\boldsymbol{GS}$ is percentage change of real government spending and other macroeconomic variables—$FDI$ (foreign direct investment), $INF$ (Inflation) $EXR$ (effective exchange rate), and $IRR$ (real interest rate); $VA$ is voice and accountability; $PS$ is political stability; $GS$ is government effectiveness; $RQ$ is regulatory quality (macro-governance factors); $a_i$ is a set of country fixed effect, $y_t$ is a year-fixed effect; and $\varepsilon_{it}$ is the error term (assumed to be distributed normally).

Nevertheless, the linear model may not take into account multifaceted effects of government spending on economic growth, as well as its asymmetric behavior or structure (Halkos and Paizanos 2013). Accordingly, the study follows empirical research on macroeconomic indicators by first developing the nonlinear quadratic formulation derived from Equation (1) to investigate the effect of government spending's asymmetric behavior on economic growth (Olaoye et al. 2020; Wu et al. 2020). Thus, the nonlinear quadratic regression model can be expressed as:

$$EG_{it} = \beta_0 + \beta_1\, \boldsymbol{GS_{it}} + \beta_2 \boldsymbol{GS_{it}^2} + \beta_3\, FDI_{it} + \beta_4\, INF_{it} + \beta_5\, EXR_{it} + \beta_6 IRR_{it}$$
$$+ \beta_7\, VA_{it} + \beta_8\, PS_{it} + \beta_9\, GS_{it} + \beta_{10}\, RQ_{it} + a_i + y_t + \varepsilon_{it} \tag{2}$$

in which $\boldsymbol{GS^2}$ is government spending squared, used to disclose the nonlinear quadratic nexus between levels of government expenditure and the economic growth rate, particularly when $GS$ and $\boldsymbol{GS^2}$ have opposite and significant signs.

Although Equation (2) can serve to test the nonlinear relation between government spending and economic growth, there is still concern about the non-monotonic nexus

between the two, which can produce positive and negative government spending shocks simultaneously and over time (Chen and Liu 2018; Pragidis et al. 2018; Atems 2019; Chen and Liu 2018; Olaoye et al. 2020). Consequently, the cubic formulation[3] of government spending can control the non-monotonic nexus effect between government spending and economic growth (Ghosh et al. 2013; Halkos and Paizanos 2013; Bökemeier and Stoian 2018). Hence, the nonlinear cubic regression model can be written as follows, developed by the nonlinear cubic formulation Equation (3) (derived from Equation (2)):

$$
\begin{aligned}
EG_{it} = \beta_0 + \beta_1 \boldsymbol{GS_{it}} + \beta_2 \boldsymbol{GS_{it}^2} + \beta_2 \boldsymbol{GS_{it}^3} + \beta_4\, FDI_{it} + \beta_5\, INF_{it} + \beta_6\, EXR_{it} \\
+ \beta_7 IRR_{it} + \beta_8\, VA_{it} + \beta_9\, PS_{it} + \beta_{10}\, GS_{it} + \beta_{11}\, RQ_{it} + a_i \\
+ y_t + \varepsilon_{it}
\end{aligned} \tag{3}
$$

in which $\boldsymbol{GS^3}$ is the cubic formulation of government expenditure, which may help identify the non-monotonic effect of positive and negative government spending shocks over time, in which a consistent sign between $GS$ and $\boldsymbol{GS^3}$, as well as an opposite sign of $\boldsymbol{GS^2}$, can confirm this non-linear cubic relation.

The POLS estimator is based upon mean values and may produce incorrect results when the data's statistical distribution includes unequal variation, in which case the nexus among the indicators can change based on dependent variables' conditional distribution. Consequently, it is necessary to use an estimator that can provide a more complete picture of the nexus among the variables (Cade and Noon 2003). Quantile regressions can address this issue by evaluating different points on the dependent variable's conditional distributions. The essential benefit of using the quantile approach is that it can address the heterogeneous structure of different government spending rules and different market conditions, while POLS can only use mean values (Allard et al. 2018). Therefore, the quantile approach can complement the POLS by dividing the dependent variable into different quantiles using the median at the 50th quantile.

Empirically, there are large differences between macroeconomic indicators' mean and median values, particularly in relation to growth (Hübler 2017), and the quantile approach is more robust to outliers compared to techniques that use mean values (Shaddady and Tomoe 2019). Hübler (2017) also stated that the quantile approach is an interesting technique to test the N-shaped hypothesis, because of the possibilities of variations in slopes across quartiles. Furthermore, the quantile estimator adopted in this study offers important benefits relative to more traditional linear methods and to smooth transition methods, including the ability to estimate the impact of explanatory variables (e.g., government spending) at different quantiles of the conditional distribution of the outcome variable. The quantile regression can also estimate vector autoregressions and associated quantile-specific impulse responses. This method is significant to estimate the effects of fiscal policy on the forecasts of various quantiles of the distribution of macroeconomic indicators as economic growth, consistent with several unique studies of fiscal policy and economic growth (Yang 2016; Linnemann and Winkler 2016). Accordingly, Equation (4) presents the quantile regression model:

$$
\begin{aligned}
Q_{\tau it}\left(EG_{\tau it}|x_t\right) = \; & a_{\tau it} + \boldsymbol{\beta_{\tau 1} GS_{it}} + \boldsymbol{\beta_{\tau 2} GS_{it}^2} + \boldsymbol{\beta_{\tau 3} GS_{it}^3} + \beta_2\, FDI + \beta_3\, INF \\
& + \beta_4\, EXR + \beta_5 IRR + \beta_6\, VA + \beta_7\, PS + \beta_8\, GS + \beta_9\, RQ + a_i \\
& + y_t + \beta_{\tau k} x_{kit} + \varepsilon_{it}
\end{aligned} \tag{4}
$$

in which $\boldsymbol{GS_{it}^2}$ and $\boldsymbol{GS_{it}^3}$ are government spending squared and government spending cubed, respectively, which are used to examine government expenditure's nonlinear effect on economic growth; $i$ is country; $t$ is time; $\tau$ reflects the quantiles (0.25, 0.50, and 0.75); and $x_{kit}$ refers to other explanatory variables.

Another econometric concern that can exist in the previous equations is bias attributable to the potential endogeneity between economic growth and government spending. Traditionally, government spending's direct or indirect influences on economic growth are assumed, but the nature of these influences varies. There is empirical and anecdotal evidence of the role economic growth plays in increased government spending, as well

as reverse causality issues related to increased taxation to finance public spending, which inhibits growth, and public spending that has long-term effects that are potentially conducive to economic growth, as discussed in other sections in more depth (Afonso and Jalles 2014). Therefore, the generalized method of moments (GMM) model can address the issue of reverse causality by assuming that government spending, economic growth, and lagged dependent variables are endogenous variables, while the other explanatory indicators are exogenous variables (Christie 2014).

Because the sample consists of panel data from different countries, there is another potential econometric issue related to the unobserved heterogeneity across countries, which can create cross-section specific error (Afonso and Jalles 2014). The fixed effects (FE) model is able to address this issue more appropriately, particularly given the possibility that countries' unobserved characteristics are correlated with economic growth, as well as other explanatory indicators (Halkos and Paizanos 2013).

### 3.2. Variables Selection

#### 3.2.1. Dependent Variable: Economic Growth

The study adopted change in real GDP growth (i.e., annual percentage growth) as a dependent variable (Table 1) (Mishchenko et al. 2018; Ferreira et al. 2020).

**Table 1.** Definitions and descriptions of the variables.

| Variables | Variable Abbreviation | Definition | Sources |
|---|---|---|---|
| *Dependent variable* | | | |
| Economic growth | EG | Change in real GDP growth as annual percentage growth | WDI |
| *Independent variables* | | | |
| Government spending | GS | Percentage change of real government spending | WDI |
| **Other macro-indicators** | | | |
| FDI | FDI | FDI inflows as percentage of GDP | WDI |
| Inflation | INF | Percentage change in consumer price index | WDI |
| Real interest rate | IRR | Lending interest rate adjusted for inflation | WDI |
| Effective exchange rate | EXR | Currency value against weighted average of several foreign currencies divided by a price deflator or index of costs | IMF |
| **Macro-governance indicators (Thomas 2010)** | | | |
| Voice and accountability | VA | The extent of citizens' freedoms of selecting their government, expression, association, and media | WGI |
| Political stability | PS | Perceptions of potential government destabilization (including political violence and terrorism) | WGI |
| Government effectiveness | GE | Quality of public services, civil service independence from political pressure, policy formulation and implementation, and government commitment to such policies | WGI |
| Regulatory quality | RQ | Ability to formulate and implement sound policies and regulation promoting competition and private sector development | WGI |

**Note 1**: Real USD is used for all monetary indicators; **Note 2**: WDI—World Development Indicators; IMF—International Monetary Fund; WGI—Worldwide Governance Indicators.

#### 3.2.2. Independent Variable: Government Spending

The study used the percentage change in real government spending as a measure of government spending's linear or nonlinear effects on economic growth, as well as to explore government spending squared and cubed, to examine the nonlinear nexus. Accordingly, the study follows previous economic literature in using the change in real government

spending as a proxy of government spending and other derived independent variables (Aizenman et al. 2019; Di Serio et al. 2020; Kronborg 2021).

### 3.2.3. Independent Variable: Macroeconomic Indicators

The study adopted four macroeconomic variables (Table 1) as a proxy for the economic environment effects controlled to determine the shape of the relation between government spending and economic growth (Pradhan et al. 2015; Gan et al. 2020). FDI is measured by FDI inflows as a percentage of GDP (Sokhanvar 2019; Asongu and Odhiambo 2020). Following previous studies (Pradhan et al. 2014, 2015), this research used the percentage change in the consumer price index, the lending interest rate adjusted for inflation, and the currency value against a weighted average of several foreign currencies as proxies for inflation rate, real interest rate, and exchange rate (respectively) (Table 1).

### 3.2.4. Independent Variable: Macro-Governance Indicators

Based upon the literature (Thomas 2010), the study used four macro-governance indicators to control the differences in governance framework across countries (Table 1). Thus, to measure democracy and freedom, the study adopted the voice and accountability indicator and used the political stability indicator for terrorism and political violence. Government effectiveness and regulatory quality indicators were applied as proxies for public services and policies' quality.

### 3.3. Data

The study used unbalanced panel data from 19 countries over the period of 1995–2019; the period 2020–2021 was excluded because of the COVID-19 pandemic's aberrant effects and missing data. The data were transformed and abstracted from three primary sources: (1) World Development Indicators (WDI), issued by the World Bank for most macroeconomic variables; (2) International Monetary Fund (IMF) for effective exchange rate; and (3) Worldwide Governance Indicators (WGI) for macro-governance variables. The study considered 19 countries, comprising most EECA nations (according to the geographical categorization of the United Nations Population Fund (UNPF): Albania; Armenia; Azerbaijan; Belarus; Bosnia and Herzegovina; Bulgaria; Georgia; Kazakhstan; Kyrgyzstan; Moldova; North Macedonia; Romania; Russia; Serbia; Tajikistan; Turkey; Turkmenistan; Ukraine; and Uzbekistan).

Table 2 reports the descriptive statistics for all variables. The mean economic growth between 1995 to 2019 was approximately 5%, ranging from −16.7% to 22.96%. Interestingly, the mean economic growth in the EECA is higher than the world average, which is approximately 3% (Table 3). This can be attributed to rapid economic reform in many countries in the EECA during their recovery from Soviet economic models from the 1990s onward. Government spending averaged approximately 15.5%, and the minimum government spending was approximately 6%, while the maximum was approximately 32%.[4] Unsurprisingly, there are high inflation levels in the EECA, averaging approximately 28%,[5] but descriptive statistics reveal an interesting outcome for the maximum value of inflation (approximately 1058%). This inflation rate was calculated in Bulgaria in 1997,[6] and dropped subsequently, while it had been lower in 1995 and 1996[7] when no large variation occurred in the average inflation (which remained steady at approximately 26.5%). It is remarkable that all macro-governance variables have average values less than 50%, which may indicate a weak macro-governance framework across most EECA countries. Nonetheless, some countries have reasonable performance in certain macro-governance elements, such as Georgia, with approximately 83% in regulation quality, and Bulgaria, with approximately 72% in political stability.

**Table 2.** Descriptive statistics of variables.

| Variable | Obs | Mean | Std. Dev. | Min | Max |
|---|---|---|---|---|---|
| EG | 475 | 4.668077 | 6.741784 | −16.7 | 22.96 |
| GS | 475 | 15.49627 | 4.258418 | 5.94 | 32.01 |
| FDI | 475 | 5.086079 | 5.58653 | −1.39 | 15.08 |
| INF | 475 | 28.02541 | 99.58989 | −8.5 | 1058.4 |
| EXR | 475 | 98.5145 | 39.69589 | 45.10725 | 476.6331 |
| IRR | 475 | 4.40016 | 13.32158 | −6.13 | 39.81 |
| VA | 475 | 32.58102 | 19.46081 | 0 | 68.26923 |
| PS | 475 | 33.7587 | 16.24975 | 3.01 | 72.51185 |
| GE | 475 | 43.58616 | 17.87952 | 1.470588 | 69.14 |
| RQ | 475 | 39.42311 | 21.3978 | 1.421801 | 83.17308 |

**Table 3.** Economic growth and government spending over time and across countries.

| Year | World-EG | EG | GS | Countries | EG | GS |
|---|---|---|---|---|---|---|
| 1995 | 3 | −0.48053 | 17.41211 | Albania | 4.602889 | 11.01 |
| 1996 | 3.5 | 6.460725 | 16.74474 | Armenia | 6.3276 | 11.14 |
| 1997 | 3.7 | 3.097102 | 17.67895 | Azerbaijan | 7.4548 | 11.7404 |
| 1998 | 2.5 | 3.568621 | 16.78579 | Belarus | 4.2484 | 17.7596 |
| 1999 | 3.25 | 2.954822 | 16.15632 | Bosnia | 9.4984 | 21.4596 |
| 2000 | 4.4 | 5.938119 | 15.86421 | Bulgaria | 2.5108 | 17.3848 |
| 2001 | 1.9 | 5.585872 | 15.94 | Georgia | 5.5324 | 12.7756 |
| 2002 | 2.17 | 5.872199 | 15.86158 | Kazakhstan | 5.012 | 11.2164 |
| 2003 | 2.95 | 6.835975 | 15.56632 | Kyrgyzstan | 4.2592 | 18.0284 |
| 2004 | 4.41 | 8.164641 | 15.29684 | Moldova | 2.9824 | 18.3988 |
| 2005 | 3.9 | 8.381406 | 15.4879 | North Macedonia | 2.649241 | 17.6788 |
| 2006 | 4.37 | 8.954362 | 15.08947 | Romania | 3.3508 | 15.0032 |
| 2007 | 4.32 | 8.881395 | 15.12632 | Russia | 2.745682 | 17.9544 |
| 2008 | 1.86 | 6.791188 | 15.08579 | Serbia | 3.402999 | 19.3352 |
| 2009 | −1.66 | −1.63492 | 16.20579 | Tajikistan | 5.4076 | 12.7832 |
| 2010 | 4.31 | 4.230363 | 15.27211 | Turkey | 4.830746 | 13.2704 |
| 2011 | 3.12 | 5.162437 | 14.63947 | Turkmenistan | 6.924 | 10.7672 |
| 2012 | 2.52 | 3.077098 | 14.77684 | Ukraine | 0.9916 | 19.4572 |
| 2013 | 2.67 | 4.651533 | 14.50947 | Uzbekistan | 5.8292 | 17.2652 |
| 2014 | 2.87 | 3.134505 | 14.51947 | **Total** | 4.668077 | 15.49627 |
| 2015 | 2.92 | 2.125984 | 14.76842 | | | |
| 2016 | 2.61 | 2.813346 | 14.91526 | | | |
| 2017 | 3.28 | 4.205301 | 14.59158 | | | |
| 2018 | 3.03 | 4.078969 | 14.47158 | | | |
| 2019 | 2.33 | 3.851417 | 14.64053 | | | |
| **Total** | 2.9692 | 4.668077 | 15.49627 | | | |

Table 3 shows the difference between the mean score of economic growth and government spending over time (left) and across counties (right). The economic growth varied over time, with negative growth during the 2009 global financial crisis; the highest growth score in 2006 was consistent with world economic growth. Overall, the region's economic growth shows reasonable levels of performance, particularly between 2005 and 2008. Conversely, the growth performance displays fluctuations after the global financial crisis. This provides a preliminary indication that the European debt crisis and oil price plunge influenced certain Eastern European[8] and Central Asian[9] countries. Government spending shows a decline over time, particularly after the global financial crisis. This may reflect a "credit-driven boom-and-bust cycle", particularly when associated with the reduction in global liquidity, which produced excessive current account deficits.

Furthermore, comparison of the mean scores for economic growth and government spending across countries presented in Table 3 shows that some oil and non-oil producing countries in Central Asia, such as Azerbaijan, Armenia, Georgia, Kazakhstan, Tajikistan, Turkmenistan, and Uzbekistan, exhibited good economic performance and reasonable levels of economic growth above the group's average. In contrast, most Eastern Europe countries have shown substantial government support, sometimes reaching approximately 20% of GDP in the case of Serbia and Ukraine, and even more in the case of Bosnia. This indicates that public debt in most Eastern Europe countries is generally lower than in more advanced European countries, which may provide flexibility for Eastern Europe governments to finance their spending. Nevertheless, increased public debt still poses considerable risks for these countries.[10]

The correlation analysis between the dependent and explanatory variables is presented in Table 4. The results indicate that multicollinearity does not appear to pose a serious issue in this analysis.[11] As can be observed, there are negative correlations between the dependent variable and most explanatory variables, except for FDI and interest rate. Notably, all macro-governance indicators show a negative correlation with dividend payment, which is an indication that a weak macro-governance framework impedes government spending. Finally, government spending exhibits a negative correlation with economic growth, which manifests prima facie evidence that government spending is detrimental to economic growth.

**Table 4.** Correlation among variables.

| Variables | (1) | (2) | (3) | (4) | (5) | (6) | (7) | (8) | (9) | (10) |
|---|---|---|---|---|---|---|---|---|---|---|
| EG (1) | 1 | | | | | | | | | |
| GS (2) | −0.1533 | 1 | | | | | | | | |
| FDI (3) | 0.1811 | −0.2245 | 1 | | | | | | | |
| INF (4) | −0.2527 | 0.0442 | −0.0523 | 1 | | | | | | |
| EXR (5) | −0.0216 | 0.2286 | −0.1002 | −0.0039 | 1 | | | | | |
| IRR (6) | 0.0324 | −0.0639 | 0.097 | −0.3002 | −0.05 | 1 | | | | |
| VA (7) | −0.1147 | 0.2519 | 0.0003 | −0.0883 | −0.2473 | 0.0681 | 1 | | | |
| PS (8) | −0.0558 | −0.0655 | 0.0708 | 0.041 | −0.082 | −0.0687 | 0.1962 | 1 | | |
| GE (9) | −0.0715 | −0.0049 | −0.0163 | −0.006 | −0.0639 | −0.018 | 0.1043 | 0.0951 | 1 | |
| RQ (10) | −0.1487 | −0.018 | 0.0256 | −0.1818 | −0.2127 | 0.1764 | 0.5627 | 0.1845 | 0.0895 | 1 |

## 4. Results

The study focused on three tests of our model: (1) estimating the parameters of Equation (1), which showed the linear relation between government spending and economic growth; (2) testing the first hypotheses derived and re-estimating the nonlinear relation between government spending and economic growth, using the nonlinear quadratic regression of Equation (2) and the U-shaped test; and (3) re-testing the hypotheses derived second on the nonlinear cubic relation between government spending and economic growth (Equation (3)). As baseline models, the study estimated the linear and nonlinear relation between government spending and economic growth by applying OLS and quantile regression (Equation (4)), the outcomes of which are displayed in Figure 1. The schemes represent all explanatory variables' distribution plots of the regression estimate for quantiles 0.25–0.75.

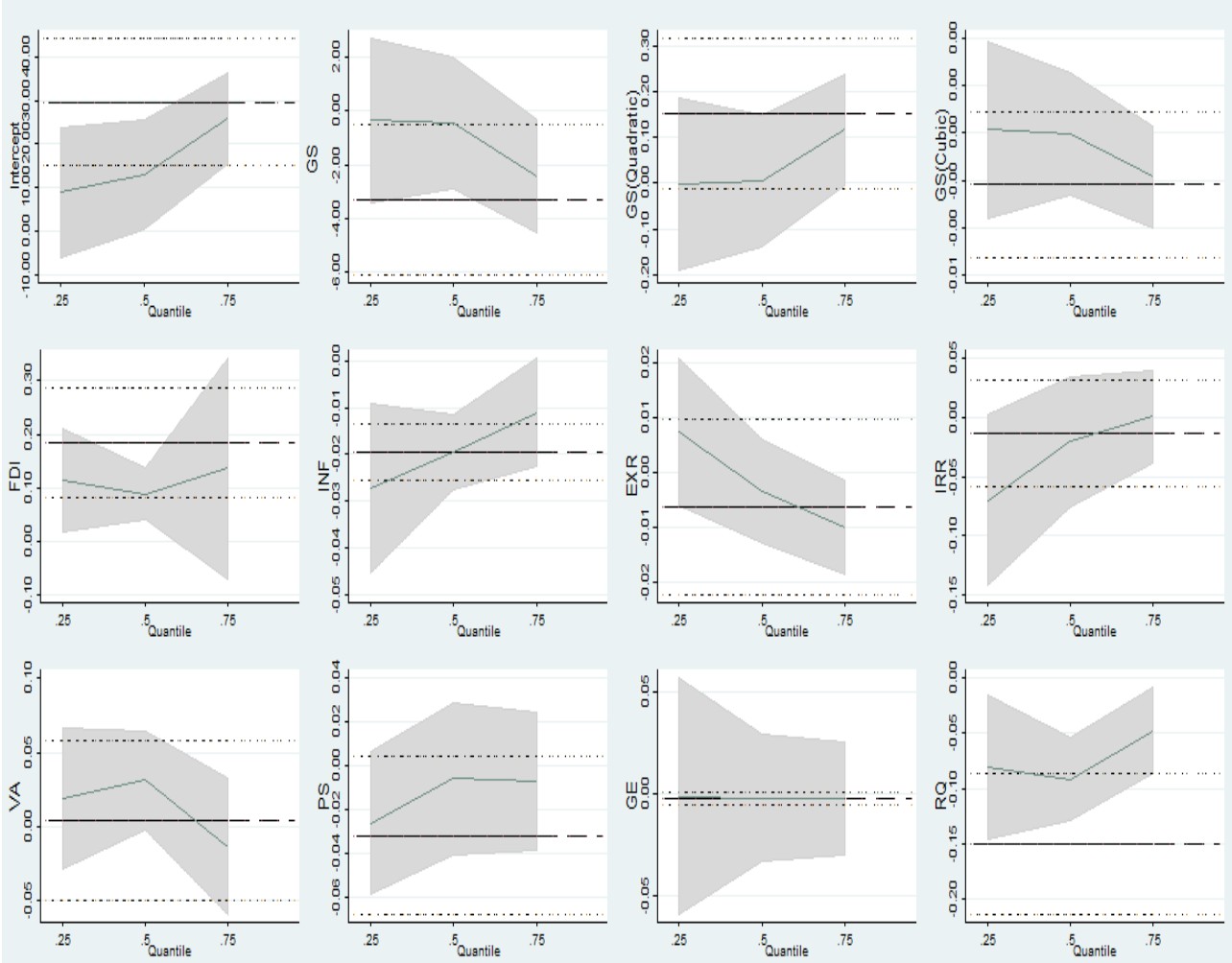

**Figure 1.** The distribution of explanatory variables at Q25, Q50, and Q75. The horizontal line represents the quantile at the 0 scale and the gray area represents the 95% confidence interval for QR. The OLS estimator is represented by the broken line.

*4.1. Relation between Government Spending and Economic Growth*

4.1.1. Linear Relation

The estimated parameters of Equation (1) are presented in Table 5, columns 1 to 4, which depict the results of the linear estimators. The outcomes indicate that the sign of the variable government spending is negative and statistically significant. This result is consistent across the two models (OLS and quantile) and remains similar in all quantiles analyzed (Q25, Q50, and Q75). Figure 1 shows a decline at Q0.5 for the distribution of GS, reflecting that governments who finance spending by borrowing and/or raising taxes inhibit economic growth in the short term. Using these financing instruments usually causes a decline in production and a commensurate decrease in real wages and consumption, and the government may crowd out the private sector, consistent with the neo-classical approach (Gemmell and Au 2013; Onifade et al. 2020). In contrast, Keynesian theory and certain empirical findings posit that public investment in infrastructure, health, and education enhances production, provides jobs, consolidates demand and financial inflows, and thus increases economic growth over the long term (Larch and Lechthaler 2013; Kimaro et al. 2017). Overall, this negative linear nexus may be taken as early evidence that government spending has a negative linear influence on economic growth; the debate between neo-classical and Keynesian theories relaxes the assumption of a linear nexus and may lead to the non-monotonic relation.

**Table 5.** Regression analysis of linear relation, nonlinear quadratic relation and nonlinear cubic relation for government spending and economic growth.

| | (1) | (2) | (3) | (4) | (5) | (6) | (7) | (8) | (9) | (10) | (11) | (12) |
|---|---|---|---|---|---|---|---|---|---|---|---|---|
| VRs | OLS | Q 25 | Q 50 | Q 75 | OLS | Q 25 | Q 50 | Q 75 | OLS | Q25 | Q50 | Q75 |
| | | | Linear | | | | | Quadratic | | | Cubic | |
| GS | −0.252 *** | −0.308 *** | −0.280 *** | −0.237 *** | −1.402 *** | −0.444 ** | −0.425 *** | −1.161 ** | −0.805 ** | −0.422 ** | −0.433 ** | −0.347 *** |
| | (0.086) | (0.0001) | (0.0020) | (0.070) | (0.5014) | (0.0017) | (0.0131) | (0.4912) | (0.169) | (0.089) | (0.035) | (0.085) |
| $GS^2$ | | | | | 0.360 *** | 0.00434 ** | 0.0483 *** | 0.0295 ** | 0.151 ** | 0.266 * | 0.00529 ** | 0.00220 ** |
| | | | | | (0.0172) | (0.0008) | (0.009) | (0.0166) | (0.077) | (0.117) | (0.0005) | (0.0001) |
| $GS^3$ | | | | | | | | | −0.00217 * | −0.00186 ** | −0.008754 *** | −0.00132 ** |
| | | | | | | | | | (0.00120) | (0.00032) | (0.0009) | (0.0006) |
| FDI | 0.191 *** | 0.111 *** | 0.906 ** | 0.237 * | 0.183 *** | 0.113 *** | 0.0878 ** | 0.144 ** | 0.184 *** | 0.136 *** | 0.879 *** | 0.113 *** |
| | (0.0625) | (0.0383) | (0.0386) | (0.142) | (0.0589) | (0.0415) | (0.0375) | (0.0642) | (0.0578) | (0.041) | (0.014) | (0.041) |
| INF | −0.0193 *** | −0.0266 *** | −0.0200 *** | −0.0408 * | −0.0199 *** | −0.0276 *** | −0.0195 *** | −0.00980 ** | −0.0195 *** | −0.0610 * | −0.0495 ** | −0.0272 *** |
| | (0.00487) | (0.00949) | (0.00750) | (0.0208) | (0.00473) | (0.00940) | (0.00723) | (0.00712) | (0.00460) | (0.0110) | (0.0100) | (0.00758) |
| EXR | −0.00391 | 0.00695 * | −0.00285 | −0.0105 *** | −0.00882 | 0.00719 * | −0.00327 | −0.00332 | −0.00322 | −0.00994 * | −0.00126 | 0.00733 * |
| | (0.00548) | (0.00187) | (0.00482) | (0.00395) | (0.00998) | (0.00173) | (0.00484) | (0.00996) | (0.00692) | (0.00290) | (0.00428) | (0.00201) |
| IRR | −0.0113 ** | −0.0663 * | −0.0476 * | −0.0835 ** | −0.0866 ** | −0.0710 * | −0.0208 ** | −0.0781 ** | −0.0138 | 0.000523 | −0.0206 | −0.0698 ** |
| | (0.0017) | (0.0198) | (0.0106) | (0.0143) | (0.0113) | (0.0207) | (0.0100) | (0.0229) | (0.0210) | (0.0220) | (0.0263) | (0.0139) |
| VA | 0.0148 | 0.0222 | 0.0327 | 0.00517 | 0.00959 | 0.0200 | 0.0313 | −0.0240 | 0.00424 | −0.0133 | 0.0313 | 0.0192 |
| | (0.0213) | (0.0359) | (0.0420) | (0.0222) | (0.0213) | (0.0365) | (0.0514) | (0.0342) | (0.0215) | (0.0339) | (0.0502) | (0.0348) |
| PS | −0.0243 | −0.0266 | −0.00222 | −0.00795 | −0.00297 | −0.00267 | −0.00567 | −0.00351 | −0.00316 | −0.00684 | −0.00563 | −0.00259 |
| | (0.0489) | (0.0392) | (0.0168) | (0.0133) | (0.0298) | (0.0199) | (0.0183) | (0.0136) | (0.0201) | (0.0600) | (0.0161) | (0.0236) |
| GE | −0.0021 *** | −0.00151 | −0.00225 | −0.00261 | −0.0029 *** | −0.00147 | −0.00222 | −0.00245 | −0.00240 *** | −0.00248 | −0.00222 | −0.00148 |
| | (0.000568) | (0.0186) | (0.0177) | (0.0276) | (0.000479) | (0.0175) | (0.0165) | (0.0241) | (0.000464) | (0.00353) | (0.00751) | (0.0223) |
| RQ | −0.172 ** | −0.0842 * | −0.0890 *** | −0.0281 | −0.154 ** | −0.0791 * | −0.0915 *** | −0.0212 | −0.151 ** | −0.0278 | −0.0915 *** | −0.0804 * |
| | (0.0739) | (0.0248) | (0.0032) | (0.0654) | (0.0661) | (0.0237) | (0.0028) | (0.0690) | (0.0653) | (0.0545) | (0.0029) | (0.0359) |
| Constant | 0.040 *** | 0.519 *** | 0.180 *** | 0.345 *** | 0.313 *** | 0.414 *** | 0.193 *** | 0.661 *** | 0.262 *** | 0.592 *** | 0.196 *** | 0.330 |
| | (0.006) | (0.064) | (0.007) | (0.016) | (0.018) | (0.047) | (0.086) | (0.028) | (0.035) | (0.057) | (0.042) | (0.715) |
| Years | Yes | Yes | Yes | Yes | Yes | Yes | Yes | Yes | Yes | Yes | Yes | Yes |
| Country | Yes | Yes | Yes | Yes | Yes | Yes | Yes | Yes | Yes | Yes | Yes | Yes |
| Obs | 475 | 475 | 475 | 475 | 475 | 475 | 475 | 475 | 475 | 475 | 475 | 475 |
| R-sq. | 0.177 | | | | 0.194 | | | | 0.197 | | | |

Table 5 presents the pooled OLS estimates for our main sample in columns 1, 5, and 9 beside quantile estimates, where quantiles are reported in columns 2–4, 6–8, and 10–12. The dependent variable is economic growth. The regressions reported in columns 1 to 4 are for linear relation. Columns from 5 to 8 report nonlinear quadratic relation. Columns 9 to 12 report nonlinear cubic relation. Robust standard errors are in parentheses; *** $p < 0.01$, ** $p < 0.05$, * $p < 0.1$. The quantiles at Q0.25, Q0.50, and Q0.75 are applied to estimate economic growth. ± F tests for the equality of the slope coefficient across various quantiles, significant at $p < 0.05$ for most quantiles; however, they are not reported to save space (the details are available upon request).

### 4.1.2. Nonlinear Quadratic Relation

To introduce the first hypothesis specification derived, the study re-estimated the base equation (Equation (1)) by including $GS^2$. The estimation of the first estimators derived (Equation (2)) is shown in Table 5 (columns 5–8). The results revealed that government spending (*GS*) has a statistically significant negative effect on economic growth, while government spending quadratic ($GS^2$) has a positive significant influence on economic growth across all quintiles and the OLS model. This significant influence of *GS* and $GS^2$ confirms a non-linear nexus, as the sign of the coefficient of $GS^2$ is positive; consequently, the non-linear nexus follows an inverse normal distribution. Figure 2 illustrates the nexus between government spending and economic growth based upon the actual regressions' outcomes in Table 5. In like manner, the distribution of $GS^2$ in Figure 1 shows an upward movement from the quantile's (Q0.50) median value.

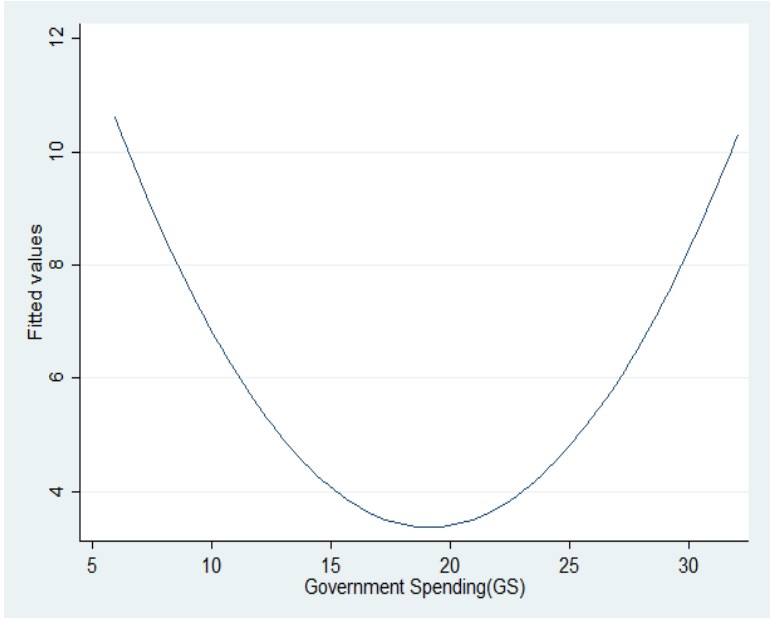

**Figure 2.** The nonlinear quadratic relation between government spending and economic growth (U-test).

This nonlinear relation indicates that the level of government spending has an inverse normally distributed inhibition effect, and then promotes economic growth, in which the inhibition attributable to disincentive effects of early shocks in rising taxes, followed by their promotion, is determined largely by increasing capital's marginal productivity and enhancing investment in core public goods. This approach is consistent with many economic theories, particularly Keynesian macroeconomic theory, which posits the essential role that government spending plays in reviving the economy. Accordingly, when the government begins to raise taxes, this has many disincentive effects on economic growth, after which enhanced spending offsets the weak pace of economic activities and controls such effects. The proponents of this school argue that government spending has a positive multiplier effect on economic growth by mitigating short-term disincentive effects that can occur when taxes are raised to finance government spending, such as fluctuations in employment and output (Zagler and Dürnecker 2003).

The nonlinear nexus displayed in Figure 2 confirms the inverse normal relation between government spending and economic growth. Initially, in financing its spending by raising taxes, the government may impede economic growth (exerting a negative influence); subsequently, after the inflection point of 19.4% of government spending (Table 6), the curve increases because of government spending's role in offsetting the weak pace in economic conditions, investing in public projects, and enhancing productivity. Nonetheless, this

outcome is inconsistent in the financial literature, which posits a valid tipping point (i.e., threshold) beyond which government spending may induce economic slowing. Numerous studies have affirmed inverse normal distribution in the government spending–economic growth nexus (Magazzino 2014; Forte and Magazzino 2016; Hajamini and Falahi 2018). Consequently, contrary findings in this and other studies (Rahn and Fox 1996; Herath 2012) suggest that government spending's effect on economic growth appears to be more profound and ambiguous than often supposed by neo-Keynesians.

**Table 6.** U-test for Government spending (GS) and Economic growth (EG).

|  | **Min** | **Max** |
|---|---|---|
| Interval | 5.94 | 32.01 |
| Slope | −0.97 *** | 0.90 *** |
|  | (−3.24) | −1.47 |
| SLM test for U shape | 1.47 *** |  |
| *p*-value | 0.07 |  |
| **Extreme point** | **19.40** |  |

Standard errors are in parentheses; *** $p < 0.01$.

### 4.1.3. Nonlinear Cubic Relation

This section presents the re-estimates of the hypotheses derived to examine the nonlinear cubic nexus between government spending and economic growth (see Equation (3)). There are several reasons to re-test the cubic nonlinear relation. First, as mentioned earlier, there is an optimal threshold level of government spending; it is assumed that if spending exceeds this level, the economy will slow. Second, macroeconomic factors can reshape the nexus between government spending and economic growth, as well as institutional infrastructure. Third, in the case of most developing countries, where governance is characterized by a lack of transparency, weak institutional framework, poor governance, and rigid structures, inadequate information about governments' fiscal operations can be assumed. Fourth, economic cycles influence the relation between government spending and economic growth through either discretionary or automatic fiscal measures (Olaoye et al. 2020).

Consequently, this study finds new evidence of the cubic nonlinear nexus between government spending and economic growth (i.e., columns 9–12 in Table 5). Specifically, government spending initially impedes economic growth (i.e., has a negative effect), and then increased government spending can enhance economic growth (i.e., has a positive effect), until the optimum levels of government spending are reached; however, further spending can subsequently hinder economic growth (exerting negative effects).

This significant effects of $GS$ (negative), $GS^2$ (positive), and $GS^3$ (negative) on economic growth are consistent across all models (see columns 9–12 in Table 5), confirming the cubic nonlinear nexus. The signs of the coefficients of $GS$, $GS^2$, and $GS^3$ are negative, positive, and negative, respectively; therefore, the cubic nonlinear relation exhibits an inverse "N-shape", as shown in Figure 3. Moreover, Figure 3 illustrates a decline in $GS$, an increase in $GS$ (quadratic), and then a decline in $GS$ (cubic) based upon the actual regressions in Table 7. The results of the nature of government spending scenarios are consistent with Figure 1, in that $GS$ declines, followed by an increase in $GS^2$, and then a subsequent decline in $GS^3$.

This result is consistent with both Keynesian macroeconomic theory and neoclassical theory. The theoretical explanation for this nexus may be that the neo-classical theory can elucidate the initial role of financing with respect to government spending with a commensurate increase in taxation and economic slowing; subsequently, Keynesian macroeconomic theory explains increased spending's benefits in increasing aggregate demand, production, and income over the longer term, which can help repair economic contractions associated with previous taxation, etc., until the benefits of spending are ultimately equal to its costs (i.e., achieve optimal levels of spending), whereupon the costs of future financing (spending) could impede the new baseline of economic growth reiteratively.

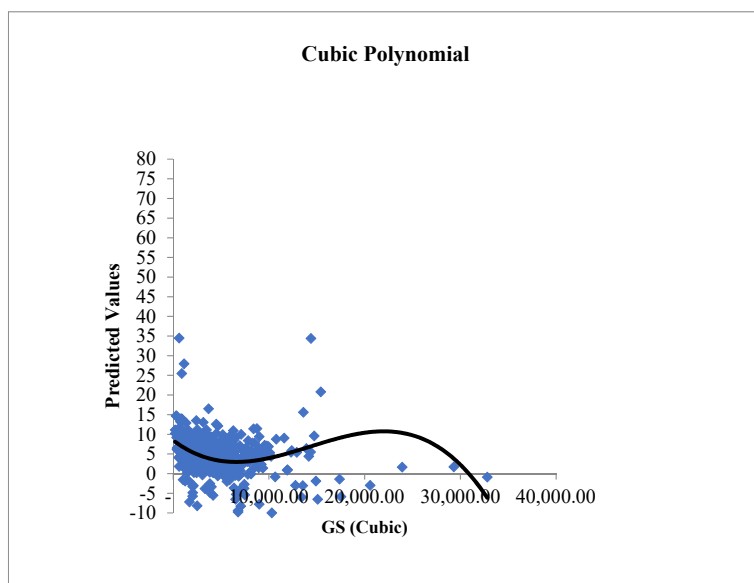

**Figure 3.** Nonlinear cubic relation between government spending and economic growth (cubic polynomial).

**Table 7.** Predicted regression of GS, $GS^2$, and $GS^3$ of cubic polynomial.

| Variables | Intercept | Coefficients | *p*-Value |
|---|---|---|---|
| GS | 13.83 | −0.62 | 0.00 |
| $GS^2$ | 191.27 | 0.22 | 0.01 |
| $GS^3$ | 2645.25 | −0.00 | 0.03 |
| $R^2$ | 0.60 | | |
| Adjusted $R^2$ | 0.54 | | |
| **Predicted (inflection point)** | **13.32** | | |
| Observations | 474 | | |

Empirically, in the case of the EECA, taxation reduces economic growth slightly, thus economies face a typical credit-driven boom-and-bust cycle, which induces increased current account deficits. A lack of global liquidity (often attributable to unpredictable factors) impedes financial inflows for EECA economies. Table 7 presents the estimation of the threshold of the cubic non-linear nexus, and predicts the optimal level of government spending, while Figure 3 provides strong evidence of the existence of an inverted "N-shaped" nexus between government spending and economic growth. The outcomes reveal that the optimal threshold level of government spending is 13.32% (i.e., the predicted value or inflection point); this is consistent with the range reported in the related literature, which has suggested a range between 11–25% (Chen and Lee 2005; Chiou-Wei et al. 2010; Altunc and Aydın 2013). In this study, the average government spending in the EECA was found to be 15.49%, which indicates that spending in most EECA economies is near the optimal level; however, some economies exceed the optimal level, such as Bosnia (21.4%), Serbia (19.3%), and Ukraine (19.4%).

*4.2. Relation between Macro-Economic and Macro-Governance Indicators and Economic Growth*

The studied macroeconomic and macro-governance indicators play key roles in determining the relation between government spending and economic growth. The outcomes show an absence of influence for most macroeconomic and macro-governance indicators in explaining economic growth, with the exceptions of inflation, FDI, and regulation quality. Although inflation is an extremely complex economic phenomenon, and there is no clear opinion on the optimal level of inflation at which growth begins to take negative paths,

government spending is considered one of the key channels for the transfer of inflationary pressures to the real economy.[12] This study's findings revealed that inflation has a significant negative influence on economic growth across all models, as shown in Table 5, which may be attributable to government spending. Furthermore, these inflationary pressures may encourage investors to prefer physical rather than financial assets, which can lead to a shortage in domestic savings. In this case, FDI can play an essential role in economic growth as the most important source of external resource flows in most EECA countries, despite the fact that these countries attract a relatively small share of the global distribution of FDI.[13] This could explain FDI's positive effect on economic growth in the EECA. Finally, the negative influence of the quality of regulation on growth can be clarified by weak levels of governance that limit states' capacity to deploy resources productively and affect their efforts to create substantive long-term returns, which may lead to increased social costs and impede long-term economic development.

*4.3. Robustness Checks: The Relation between Government Spending and Economic Growth for Grouped Variables*

Several studies have provided rationales for the nonlinear nexus between government spending and economic growth, particularly with respect to the roles of macroeconomic factors and governance (Olaoye et al. 2020). Table 8 groups the variables into three categories ((1) government spending indicators; (2) macroeconomic indicators; and (3) macro-governance indicators), and then examines macroeconomic indicators and macro-governance indicators' roles individually to determine the ways they influence the relation between government spending indicators and economic growth. As mentioned earlier, the nexus between the two can depend upon macroeconomic indicators via structural breaks or macroeconomic shocks (Kim et al. 2018). In like manner, macro-governance can influence the quality of institutional infrastructure and quality of information about government spending (Paleologou 2013). The results in Table 8 reveal that the $GS$, $GS^2$, and $GS^3$ variables are robust and affect growth significantly across all models.

*4.4. Robustness Checks: Generalized Method of Moments, Fixed Effect, and Instrumental Variable (Fitted Values)*

Wagner's law of public expenditure states that government spending tends to increase with higher levels of per capita GDP, positing that economic development increases public spending; consequently, the possibility of a reverse causality between government spending and economic growth needs to be considered, which was undertaken in this study using dynamic GMM, as shown in Table 9 (columns 1, 2, and 3). It can be seen that the results are robust and consistent across all models. However, there are still some problematic features in the sample, such as country-specification fixed effects; hence, the paper relies on a fixed effect estimator (FE) to control any possibility of both time-invariant individual country characteristics and time fixed effects. Finally, the study adopts the instrumental variable (IV) by using two-stage least squares (2SLS) regression estimator, in order to address any possibility of an endogeneity problem between government spending and economic growth, particularly to investigate economic development's causal effect on government expenditures. Consequently, government spending is considered the dependent variable in the first stage of the estimation using the instrumental independent variables: (1) fixed capital formation; (2) life expectancy at birth; (3) urbanization; and (4) official development. The outcomes of the first stage can then be used as independent variables in the second stage, together with predicted government spending (the fitted value) to estimate the nexus between government spending (fitted value, i.e., predicted value) and economic growth. The results presented in Table 9 for all three robustness models (GMM, FE, and IV) support a non-linear cubic relation between government spending and economic growth.

**Table 8.** Grouped regression analysis of governments spending and economic growth.

| | (1) | (2) | (3) | (4) | (5) | (6) | (7) | (8) | (9) | (10) | (11) | (12) |
|---|---|---|---|---|---|---|---|---|---|---|---|---|
| **Variables** | **OLS** | **Q25** | **Q50** | **Q75** | **OLS** | **Q25** | **Q50** | **Q75** | **OLS** | **Q25** | **Q50** | **Q75** |
| *Government spending indicators* | | | | | | | | | | | | |
| GS | −0.537 *** | −0.517 *** | −0.253 *** | −0.065 ** | −0.304 *** | −0.971 *** | −0.615 ** | −0.307 *** | −0.572 *** | −0.096 * | −0.629 ** | −0.285 * |
| $GS^2$ | 0.220 *** | 0.196 ** | 0.168 ** | 0.152 * | 0.206 *** | 0.157 ** | 0.126 *** | 0.164 *** | 0.164 ** | 0.111 * | 0.0804 *** | 0.167 ** |
| $GS^3$ | −0.00337 *** | −0.00374 ** | −0.00287 ** | −0.00259 ** | −0.0030 *** | −0.00285 *** | −0.00203 * | −0.00269 ** | −0.00243 ** | −0.00222 ** | −0.0015 ** | −0.00291 * |
| *Macro-economic indicators* | | | | | | | | | | | | |
| FDI | | | | | 0.166 *** | 0.139 *** | 0.1000 *** | 0.0584 * | | | | |
| INF | | | | | −0.0185 *** | −0.0227 *** | −0.0201 *** | −0.00819 ** | | | | |
| REX | | | | | −0.00108 | 0.0114 ** | 0.00321 | −0.00202 | | | | |
| IRR | | | | | −0.0312 | −0.0649 | −0.0374 | −0.00498 | | | | |
| *Macro-governance indicators* | | | | | | | | | | | | |
| VC | | | | | | | | | 0.0361 | 0.0312 | 0.0349 | 0.0065 |
| PS | | | | | | | | | −0.0257 | −0.0139 | −0.0165 | −0.00073 |
| GE | | | | | | | | | −0.00195 *** | −0.00147 | −0.00212 | −0.0026 |
| RQ | | | | | | | | | −0.0532 ** | −0.0461 | −0.0653 ** | −0.0452 * |
| Constant | 0.056 *** | 0.333 *** | 0.603 *** | 0.732 *** | 0.399 *** | 0.969 ** | 0.255 *** | 0.878 *** | 0.359 *** | 0.170 * | 0.840 *** | 0.014 *** |
| | 0.428 | 0.882 | 0.093 | 0.983 | 0.251 | 0.785 | 0.014 | 0.025 | 0.551 | 0.459 | 0.422 | 0.993 |
| Observations | 475 | 475 | 475 | 475 | 475 | 475 | 475 | 475 | 475 | 475 | 475 | 475 |
| R-squared | 0.059 | | | | 0.147 | | | | 0.078 | | | |

This table presents the pooled OLS estimates and quantile estimates. Columns 1 to 4 report linear, nonlinear quadratic, and nonlinear cubic relations of quantiles at Q0.25, Q0.50, and Q0.75, to estimate economic growth independently. Columns 5 to 8 consider macroeconomic indicators. Columns 9 to 12 show macro-governance indicators. Robust standard errors are in parentheses; *** $p < 0.01$, ** $p < 0.05$, * $p < 0.1$. The quantiles at Q0.25, Q0.50, and Q0.75 are applied to estimate economic growth. $\pm$ F tests for the equality of the slope coefficient across various quantiles, significant at $p < 0.05$ for most quantiles; however, they are not reported to save space (the details are available upon request).

**Table 9.** GMM model, fixed effect model, and instrumental variable 2-stage regressions for non-linear relationship between government spending and economic growth.

| Variables | (1) GMM | (2) GMM | (3) GMM | (4) FE | (5) FE | (6) FE | (7) IV | (8) IV | (9) IV |
|---|---|---|---|---|---|---|---|---|---|
| $EG_{t-1}$ | 0.234 *** | 0.255 *** | 0.268 *** | | | | | | |
| | (0.085) | (0.084) | (0.084) | | | | | | |
| GS | **−0.344 ***** | −0.811 * | −0.0721 ** | **−0.266**** | −1.266 * | −2.473 *** | | | |
| | **(0.012)** | (0.178) | (0.017) | **(0.013)** | (0.629) | (1.993) | | | |
| $GS^2$ | | **0.0143 **** | 0.360 *** | | **0.0294 ***** | 0.604 ** | | | |
| | | **(0.0015)** | (0.011) | | **(0.0026)** | (0.111) | | | |
| $GS^3$ | | | **−0.0990 **** | | | **−0.0140 **** | | | |
| | | | **(0.0022)** | | | **(0.0019)** | | | |
| GS (fitted value) | | | | | | | **−0.484 **** | −0.559 *** | −0.897 *** |
| | | | | | | | **(0.015)** | (0.021) | (0.070) |
| $GS^2$ (fitted value) | | | | | | | | **0.446 ***** | 0.920 *** |
| | | | | | | | | **(0.0766)** | (0.047) |
| $GS^3$ (fitted value) | | | | | | | | | **−0.0554 ***** |
| | | | | | | | | | **(0.0015)** |
| FDI | 0.128 ** | 0.127 ** | 0.127 ** | 0.188 *** | 0.198 *** | 0.112 *** | 0.167 ** | 0.0802 * | 0.0670 ** |
| | (0.0547) | (0.0540) | (0.0546) | (0.0372) | (0.039) | (0.0393) | (0.0654) | (0.0344) | (0.0334) |
| INF | −0.0167 ** | −0.0171 *** | −0.0170 *** | −0.0198 *** | −0.0200 *** | −0.0198 *** | −0.0187 *** | −0.0245 *** | −0.0217 *** |
| | (0.0017) | (0.0017) | (0.0016) | (0.00655) | (0.00645) | (0.00635) | (0.00519) | (0.00507) | (0.00505) |
| EXR | −0.0597 * | −0.0585 | −0.0584 | −0.000534 | −0.0041 | −0.00323 | −0.00103 | −0.0029 | −0.00109 |
| | (0.0265) | (0.0766) | (0.0672) | (0.00468) | (0.00714) | (0.00711) | (0.00859) | (0.00832) | (0.00826) |
| RRI | −0.0172 | −0.0201 | −0.0202 | −0.0318 | −0.0400 ** | −0.0287 | −0.0207 | −0.0844 ** | −0.0760 * |
| | (0.0552) | (0.0528) | (0.0520) | (0.03440) | (0.0220) | (0.03110) | (0.03970) | (0.0395) | (0.0389) |
| VA | 0.0615 | 0.082 | 0.0803 | 0.0318 | 0.0275 | 0.0279 | 0.021 | 0.0247 | 0.00613 |
| | (0.0959) | (0.0972) | (0.0976) | (0.0469) | (0.0444) | (0.0438) | (0.0358) | (0.0294) | (0.0294) |
| PS | 0.00473 | 0.00625 | 0.0063 | 0.00146 | 0.00531 | 0.00576 | 0.00131 | 0.00554 | 0.00199 |
| | (0.0254) | (0.0258) | (0.0260) | (0.0239) | (0.0241) | (0.0241) | (0.0255) | (0.0244) | (0.0243) |
| GE | −0.000836 | −0.008304 * | −0.000805 | −0.00295 ** | −0.00296 * | −0.0391 *** | −0.00325 * | −0.00286 * | −0.00225 |
| | (0.00594) | (0.00543) | (0.00659) | (0.00175) | (0.00107) | (0.00221) | (0.00177) | (0.0017) | (0.00368) |
| RQ | −0.121 *** | −0.120 *** | −0.123 *** | −0.738 * | −0.124 *** | −0.420 * | −0.815 *** | −0.0468 * | −0.0637 ** |
| | (0.0370) | (0.0380) | (0.0368) | (0.288) | (0.0745) | (0.144) | (0.0257) | (0.017) | (0.0270) |

**Table 9.** *Cont.*

| Variables | (1) GMM | (2) GMM | (3) GMM | (4) FE | (5) FE | (6) FE | (7) IV | (8) IV | (9) IV |
|---|---|---|---|---|---|---|---|---|---|
| Constant | 0.574 *** | 0.919 *** | 0.481 ** | 0.582 *** | 0.672 *** | 0.376 ** | 0.124 * | 0.221 *** | 0.266 *** |
| Time | Yes | Yes | Yes | Yes | Yes | Yes | Yes | Yes | Yes |
| Countries | Yes | Yes | Yes | Yes | Yes | Yes | Yes | Yes | Yes |
| AB test AR (1) | 0.006 | 0.003 | 0.004 | | | | | | |
| AB test AR (2) | 0.580 | 0.655 | 0.632 | | | | | | |
| Hansen test (p-value | 0.321 | 0.291 | 0.355 | | | | | | |
| Instruments | 28 | 33 | 34 | | | | | | |
| Groups | 41 | 41 | 41 | | | | | | |
| R-squared | | | | 0.551 | 0.461 | 0.562 | 0.691 | 0.575 | 0.204 |
| Observations | 437 | 437 | 437 | 475 | 475 | 475 | 475 | 475 | 475 |

This table presents the GMM, fixed effects (FE), and instrumental variables (IV) models. Columns 1 to 3 report linear, nonlinear quadratic, and nonlinear cubic of GMM. Columns 4 to 6 present fixed effects. Columns 7 to 9 show instrumental variables. Robust standard errors are in parentheses; *** $p < 0.01$, ** $p < 0.05$, * $p < 0.1$. The Hansen test is a test to over-identify restrictions in the GMM model. AB test AR (1) and AR (2) refer to the Arellano–Bond test for average autocovariance in residuals of order 1 and order 2, respectively: 0 (H0: no autocorrelation); *p*-values in parentheses. **Note:** Fitted value is the predicted value of $GS$, $GS^2$, and $GS^3$ based upon the first stage regression.

### 5. Conclusions

This study examined the nonlinear relation in the nexus between government spending and economic growth using unbalanced panel data from 19 countries in the EECA over the period of 1995–2019. The evidence suggests that government spending levels have a contrasting effect on economic growth (i.e., a non-monotonic relation with it). To interpret the outcomes, the study concentrated on the estimates of OLS and quantile estimators provided, because the latter can control the heterogeneity problem by estimating the parameters of each of the dependent variable's quantiles, and estimates are robust with the GMM, IV, and FE estimators.

The preliminary findings revealed that government spending has a negative effect on economic growth; nevertheless, the nonlinear quadratic estimator shows both negative and positive effects arising from the influence of government spending on economic growth, which exhibits an inverse normal distribution. The results of cubic nonlinear estimation supported a non-monotonic nexus between government spending and economic growth, thus providing strong evidence of the existence of an inverted N-shape, because of the typical credit-driven boom-and-bust cycle in most EECA countries. The outcomes revealed that the optimal threshold level of government spending (i.e., predicted value or inflection point) is 13.32%; beyond this level, more government spending exerts a negative effect on economic growth.

The findings of this study offer certain recommendations for policymakers, investors, managers, and shareholders. First, the study discussed the non-monotonic effect between government spending and economic growth and the negative effects of the nature of this relationship on fiscal policy. Consequently, policymakers and governments should consider government spending's non-monotonic effect on economic growth, particularly the negative effects of spending on private consumption and growth, and also on the other economic factors such as inflation, interest rates, and exchange rates.

Second, the study found that ill-considered government spending may weaken the role of the private sector in economic growth. Although government intervention might achieve some economic enhancements in certain developing countries, increased state expenditure that crowds out the private sector can ultimately impede private sector development. Hence, there is a need for private–public partnerships to finance public infrastructure.

Third, the study revealed the responsibility of government credit in promoting or impeding economic growth (i.e., the credit-driven boom-and-bust cycle in most EECA countries). Policy makers can establish safe levels of government credit that support stable and sustained economic growth over longer periods.

Finally, the study finds that there is a need to invest in macro-governance elements. Government efforts to promote economic development must broaden their focus from a narrow traditional economic concern with expenditure and concentrate on macroeconomic stability in addition to enhanced governance, justice, and external defense. Furthermore, governments need to focus on other provisions of such public goods as clean environmental conditions, dispute resolution, social stability, poverty alleviation, and defense of human rights.

**Funding:** This research received no external funding.

**Institutional Review Board Statement:** Not applicable.

**Informed Consent Statement:** Not applicable.

**Data Availability Statement:** Data sharing is not applicable to this article.

**Conflicts of Interest:** The author whose name is listed immediately below certifies that he has NO affiliation with or involvement in any organization or entity with any financial interest (such as honoraria; educational grants; participation in speakers' bureaus; membership, employment, consultancies, stock ownership, or other equity interest; and expert testimony or patent-licensing arrangements), or non-financial interest (such as personal or professional relationships, affiliations, knowledge or beliefs) in the subject matter or materials discussed in this manuscript.

## Notes

1. According to an OECD iLibrary report "... this region has long been at a global crossroads—at the intersection of diverse cultures, trade routes and relations, political systems." Thus, many international organisations have considered EECA countries as one panel that share educational, health, economic and political issues; see World Bank, OECD, World Health Organization.

2. See World Bank Country and Lending Groups report.

3. −16.7% in Tajikistan in 1996 during the Tajikistani Civil War; 22.96% in Bosnia in 1996, perhaps because of a boom after the Bosnia War (1992–1995).

4. Government spending was 6% in Turkmenistan, a desert country, in 1996, before oil and gas exploitation, and it was 32% in Uzbekistan in 2015, associated with high government revenues from oil, gas, and gold.

5. Nearly all countries in the EECA recorded high inflation levels, particularly in the mid-1990s.

6. A financial crisis led to Bulgaria's hyperinflation in 1997 (Charles and Marie 2017).

7. To assess whether certain high values may influence the average.

8. Many Eastern European nationals work in EU countries, and a considerable number lost their employment and returned home during the EU debt crisis and other crises (Esposito et al. 2014).

9. Central Asian countries are the main oil and gas producers in the region.

10. Most Eastern Europe countries' average public debt increased from 26% at the end of 2007 to 54% at the end of 2020. However, there is still variation from one country to another (Semik and Zimmermann 2021).

11. A correlation coefficient exceeding (0.7) indicates a potential problem (Anderson et al. 2016).

12. The fight against inflation continues throughout Central and Eastern Europe. ING.

13. The average FDI did not exceed 5%; see Table 2.

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
