# Peer review of "Is Government Spending an Important Factor in Economic Growth? Nonlinear Cubic Quantile Nexus from Eastern Europe and Central Asia (EECA)"

_economies, doi:10.3390/economies10110286_

Round 1

Reviewer 1 Report

This paper analyzes the impact of government spending on GDP growth rate in EECA countries. I think this paper investigates an important research topic. However, this paper has shortcomings in several aspects. I state my detailed comments and suggestions as below. I hope they can help the authors improve the paper in the future.

Comments and Suggestions:

1. The statistical significance levels reported in tables are not consistent with that implied by the values of standard errors. Some things are wrong. For example, in column (1) of Table 5, it is reported that the estimated coefficient of GS is statistically significant at 1% level. However, the standard error associated with the estimated coefficient is 0.3863, which is larger than the absolute value of coefficient -0.252. Obviously, the coefficient is not significant. There are many analogous errors in Table 5 and Table 9.

2. The year- and country-fixed effects are not included in Equations (1) – (3). However, in Table 5, it is stated that “Years Yes Yes Yes …” and “Country Yes Yes Yes …”. Some revisions should be made.

3. The definition of the variable GS is unclear. In Lines 359-360 on Page 8, it is stated that “the study used the percentage change in real government spending as a measure of government spending”. However, in Line 281 on Page 6, it is stated that “GS is government spending as a percentage of GDP”. These two statements are not consistent.

4. There are some grammatical mistakes and typos. Let my just give two examples here. (1) Title: “Does” should be “Is”. (2) Line 6, Page 1: “19 counties” should be “19 countries”.

Author Response

Special thanks to you for your good comments. We appreciate for Editors/Reviewers' warm work earnestly, and hope that the correction will meet with approval.
 If you have any question about this paper, please feel free to contact us. 

Once again, thank you very much for your comments and suggestions.

Kindly, see the attachment

Reviewer 2 Report

"Dose Government Spending an Important Factor in Economic 2

Growth? Nonlinear Cubic Quantile Nexus from Eastern Europe 3

and Central Asia (EECA) "

My criticisms of the manuscript are as follows:

- The reason for the country selection in the study is based on. It should be explained in detail.

- The reasons for the preferred model should be explained in detail.

- In the literature search section, the publications of authors such as Seyfettin ErdoÄŸan, Muhammad Shahbaz, Recep Ulucak should be used.

- The policy recommendations section should be expanded in line with the findings.

Author Response

(The authors gave the same response as above.)

Round 2

Reviewer 1 Report

The authors have partially revised the article based on my previous comments in the first-round review report. I appreciate their effort. However, after I carefully read this revised version, I find that three problems mentioned in my previous review report still exist. I think the paper should be revised again.

Comments and Suggestions:

1. The statistical significance levels reported in tables are not consistent with that implied by the values of standard errors. Some things are wrong. For example, in column (10) of Table 5, it is reported that the estimated coefficient of GS^2 is statistically significant at 10% level. However, the coefficient is 0.116 and the standard error associated with the estimated coefficient is 0.017, indicating that the significance level should not be just 10%. There are many analogous errors in Table 5 and Table 9.

2. The year-fixed effects are not included in Equations (1) – (3). However, in Table 5, it is stated that “Years Yes Yes Yes …”. Some revisions should be made. By the way, the subscript “it” should be added to every explanatory variable in Equations (1) – (3).

3. There are still some typos in the text. Let my just give one example here. In the abstract, “19 counties” should be “19 countries”.

Author Response

General feedbacks:

We have improved and developed the methodology, results and conclusion.

  • The statistical significance levels reported in tables are not consistent with that implied by the values of standard errors.

We have re-estimated results and re-corrected some typos error, see Tables 5 and 9.

  • The year-fixed effects are not included in Equations (1) – (3). However, in Table 5, it is stated that “Years Yes Yes Yes …”. Some revisions should be made. By the way, the subscript “it” should be added to every explanatory variable in Equations (1) – (3).

We added year-fixed effects thus  is a set of year-fixed effects in equations 1,2,3, also, we added all explanatory variable in Equations (1) – (3).

  • There are still some typos in the text. Let my just give one example here. In the abstract, “19 counties” should be “19 countries”.

We sent the paper again for professional proofreader. 

Round 3

Reviewer 1 Report

The authors have partially revised the article based on my previous comments in the second-round review report. I appreciate their effort. However, after I carefully read this revised version, I find that some problems mentioned in my previous review report still exist. I think the paper should be revised again.

Comments and Suggestions:

1. The statistical significance levels reported in tables are not consistent with that implied by the values of standard errors. Some things are wrong. For example, in column (7) of Table 5, it is reported that the estimated coefficient of Constant is statistically significant at 1% level. However, the coefficient is 0.193 and the standard error associated with the estimated coefficient is 0.636, indicating that the significance level should not be 1%. There are many analogous errors in Table 5 and Table 9.

2. The subscript “it” should be added to every explanatory variable in Equations (1) – (3).

3. There are still some typos and grammar mistakes in the text. Let my just give one example here. In the paragraph under Equation (1), “y_t is a year-fixed effects” is not grammatically correct because the word “effects” is plural. I suggest the authors to carefully read and examine the whole paper by themselves.

Author Response

General feedbacks:

We have improved and developed the methodology, results and conclusion.

  1. The statistical significance levels reported in tables are not consistent with that implied by the values of standard errors.Some things are wrong. For example, in column (7) of Table 5, it is reported that the estimated coefficient of Constant is statistically significant at 1% level. However, the coefficient is 0.193 and the standard error associated with the estimated coefficient is 0.636, indicating that the significance level should not be 1%. There are many analogous errors in Table 5 and Table 9.

We have re-estimated results and re-corrected some typos error, see Tables 5 and 9.

  1. The subscript “it” should be added to every explanatory variable in Equations (1) – (3).

We added "it", thus   will add for all explanatory variable in Equations (1) – (3).

  1. There are still some typos and grammar mistakes in the text.Let my just give one example here. In the paragraph under Equation (1), “y_t is a year-fixed effects” is not grammatically correct because the word “effects” is plural. I suggest the authors to carefully read and examine the whole paper by themselves.

We sent the paper again for professional proofreader and we got a proofreading certificate

Round 4

Reviewer 1 Report

The authors have partially revised the article based on my previous comments in the third-round review report. I appreciate their effort. However, after I carefully read this revised version, I find that a problem mentioned in my previous review report still exists. I think the paper should be revised again.

Comments and Suggestions:

Some of the statistical significance levels reported in tables are not consistent with that implied by the values of standard errors. Some things are wrong. For example, in column (1) of Table 5, it is reported that the estimated coefficient of Constant is statistically significant at 1% level. However, the coefficient is 0.040 and the standard error associated with the estimated coefficient is 0.060, indicating that the significance level should not be 1%. There are some analogous errors in Table 5 and Table 9.

I have repeatedly talked about this “significance level” issue in my first-, second- and third-round review reports. I am quite confused about why the authors have not dealt with this problem completely in the previous rounds of revisions. In every new version of the paper, some new values were presented in the tables. This is indeed strange.

Author Response

Report

Reviewer 1

General feedbacks:

We have improved and developed the methodology, results and conclusion.

Some of the statistical significance levels reported in tables are not consistent with that implied by the values of standard errors.

Hi

Thank you for your recommendations

We have re-estimated results and re-correct any errors Now, all results in Table 5 and 9 are correct and coefficient is consistent with standard error

Round 5

Reviewer 1 Report

The authors have made some revisions. I have no further question.